# Poly(oxyethylene)/Poly(oxypropylene) butyl ether prolongs the repellent effect of N,N-diethyl-3-toluamide on the skin

**Mayu Kawaguchi**[1], **Kana Matsumoto**[2], **Joji Yoshitomi**[2], **Hiroko Otake**[2], **Kanta Sato**[2], **Atsushi Taga**[2], **Tatsuji Sasabe**[1], **Kenji Nobuhara**[1], **Akira Matsubara**[1], **Noriaki Nagai**[2]*

**1** Earth Corporation, Sakoshi, Ako, Hyogo, Japan, **2** Faculty of Pharmacy, Kindai University, Kowakae, Higashi-Osaka, Osaka, Japan

* nagai_n@phar.kindai.ac.jp

**Data Availability Statement:** All relevant data are within the paper and its Supporting Information files.

## Abstract

N,N-diethyl-meta-toluamide (DEET) is a widely used insect repellent, with minimal skin permeation and sustained repellent activity in the superficial layers of the skin. In this study, we prepared a 10% DEET formulation consisting of 40% ethanol with or without 2% poly(oxyethylene)/poly(oxypropylene) butyl ether (POE-POP), an amphiphilic random copolymer. Further, we demonstrated the effects of POE-POP on tensile stress (stickiness), hydrophobicity, skin retention, permeation, and repellent activity of DEET. Stickiness was measured in male ICR mice (7-week old), and skin retention and permeation were evaluated in male Wistar rats (7-week old). In addition, female *Aedes albopictus* were used to measure the repellent action of DEET. The addition of POE-POP did not affect stickiness, volatility, and degradability but decreased logP and increased viscosity of DEET. Next, we demonstrated the behavior of DEET formulations in the rat skin. POE-POP prolonged the retention of DEET in the superficial layers of the rat skin (skin surface and stratum corneum) and decreased the penetration of DEET into rat skin tissues (epithelium and dermis). The repellent effect of DEET was also enhanced by the addition of POE-POP. However, severe skin damage was not observed after repetitive treatment with DEET formulations containing POE-POP for one month (twice a day). In conclusion, we demonstrated that a 10% DEET formulation consisting of 40% ethanol and 2% POE-POP attenuated the skin penetration and prolonged the repellent action of DEET without causing stickiness and skin damage. We conclude that the combination of ethanol and POE-POP is useful as a safe and effective delivery system for the development of insect repellent formulations containing DEET.

## Introduction

Repellents, mainly topical formulations, are generally used for protection against mosquito-borne diseases; 1-(1-ethylpropoxycarbonyl)-2-(2-hydroxyethyl) (picaridin or icaridin), lemon eucalyptus essential oil, neem, N,N-diethyl-meta-toluamide (DEET), citronella, and ethyl butylacetylaminopropionate (IR3535) are used as repellents [1,2]. Among these, DEET is the most efficient repellent and considered as the reference repellent by the World Health Organization

**Funding:** The authors received no specific funding for this work.

**Competing interests:** The authors have declared that no competing interests exist.

[3]. The mechanism of action involves binding of DEET to olfactory receptors of mosquitoes, thus repelling them from their target still in flight, and to chemoreceptors that would suppress the mosquito's biting behavior upon contact with the skin [4,5]. Low cost, non-toxic, avoiding entry into the bloodstream, effective for at least 8 h, absence of skin irritation, ability to repel the largest number of species simultaneously, water and abrasion resistance, and odorless are the ideal characteristics of insect repellents [6]. DEET topical formulations contain an allowable DEET concentration of up to 30%; the formulations containing 7–10% and 20–30% DEET are used for short periods (approximately 2 h) and longer periods (approximately 6 h), respectively [7]. However, because of its lipophilic nature, DEET can be absorbed by the skin and pass through the cutaneous barrier, reaching deeper layers of the skin and blood vessels [8]. Several studied have reported toxic effects of DEET such as skin rash, seizures, encephalopathy, and central nervous system toxicity [9–11]. Therefore, there is a need to develop topical formulations with minimum permeation of DEET and retention of its repellent activity [12].

The regulation of DEET behavior in the skin is important to utilize the advantages of DEET. Drug encapsulation in micro- and nanoparticles, drug loading into a proper absorbent system, oily and volatile compounds, polymer blends, liposomes, nanoemulsions, and cyclodextrin complexes have been introduced as interesting strategies for the development of safe and effective repellent formulations [7,13–17]. A previous study showed that Spermaceti® and Polawax®-based solid lipid microparticles encapsulating 15% DEET were safe due to reduced skin permeation along with retention of repellent action [18]. In addition, 15% Pluronic F127 micellar gel containing 10% DEET enhanced skin retention and drastically reduced skin permeation of the repellent [8]. We designed a DEET formulation using a combination of ethanol (EtOH) and 0.1% cyclodextrin as solvents, and observed reduced drug transdermal penetration and prolonged repellent effect compared with that of DEET formulation without cyclodextrin [19]. Thus, changes in the hydrophobicity and stabilization of DEET using polymers and cyclodextrin are expected to be promising strategies for DEET delivery.

Polyoxyethylene (POE) alkyl ether-type nonionic surfactants are generally non-irritant, non-toxic, and safe [20], and the hydrophilic/lipophilic balance of POE can be easily adjusted by altering the alkyl group and POE chain length. Furthermore, a polyoxypropylene (POP) chain is introduced between the POE chain and the alkyl chain of POE-type surfactant to obtain poly(oxyethylene)/poly(oxypropylene) alkyl ether-type nonionic surfactant. POP polymers are used as solubilizers and additives in microemulsion production in industries. In addition, these amphiphilic copolymers have potential advantages of remarkable design flexibility for controlling functionality and nanostructure compared with lipids and low-molar mass surfactants [21–23]. In this study, we prepared DEET formulations using poly(oxyethylene)/poly(oxypropylene) butyl ether (POE-POP), an amphiphilic random copolymer containing POE and POP groups. Moreover, we demonstrated the effects of POE-POP on the hydrophobicity, skin retention, permeation, and repellent action of DEET.

## Materials and methods

### Chemicals

All chemicals used were of the highest degree of purity commercially available. DEET and poly(oxyethylene)/poly(oxypropylene) butyl ether (UNILUBE® 50MB-11, POE-POP) were obtained from Combi-Blockes Inc. (San Diego, CA, USA) and NOF CORPORATION (Tokyo, Japan), respectively. The O.C.T. compound was provided by Sakura Finetek Japan Co., Ltd. (Tokyo, Japan). DEET (10%) was added to purified water containing 40% EtOH (DEET-E), and DEET formulations with POE-POP were prepared by the addition of 2% POE-POP to DEET-E (DEET-EPP).

## Animals

All animal experiments were performed according to the guidelines of Kindai University and Japanese Pharmacological Society. The types and doses of anesthetic agents used for anesthesia and euthanasia of mice and rats were determined according to AVMA Guidelines for the Euthanasia of Animals: 2020 Edition. In this study, animal protocols were approved by Kindai University (project codes: KAPS-31-001 and KAPS-31-010; date of approval: April 1, 2019). Wistar rats and ICR mice (male, 7-week old) were provided by Kiwa Laboratory Animals Co. Ltd. (Wakayama, Japan). These animals were raised under standard conditions [12 h/day fluorescent light (07:00–19:00), 25°C] with free access to water and CE-2 diet (Clea Japan Inc., Tokyo, Japan). The mosquitoes used in this study were *Aedes albopictus* (female, obtained from Nagasaki University, 7–14 days after emergence), which were reared in 25×25×25 cm metal mesh cages (50 female mosquitoes per cage) in rooms maintained at 28°C, 60% relative humidity, and a light:dark cycle of 16 h:8 h. Mosquitoes had free access to 5% sugar water (Nissin Sugar Co., Ltd., Tokyo, Japan). Observations of animal behavior were performed daily.

## Tensile stress of DEET formulations

The tensile stress of ICR mice was measured as described in our previous study [19]. Briefly, the dorsal and abdominal hair of the mice were removed. After one day, the mice were euthanized by injecting a lethal dose (200 mg/kg) of pentobarbital, and 25 $cm^2$ (5×5 cm) skin samples were collected. After that, the abdominal skin was treated with the formulations (1.67 μL/$cm^2$) containing DEET at 22°C.; DEET-treated skin was overlapped with the other skin (dorsal skin), which was peeled off using Force Tester MCT-2150 (A & D Co., LTD., Osaka, Japan), and the force was expressed as tensile stress (stickiness).

## Volatility of DEET formulations

The volatility of DEET formulations was measured according to a previous study [19]. DEET formulations were applied to absorbent cotton, and the weight of cotton with or without DEET formulations was measured. The absorbent cotton with DEET formulation was then stored at 25°C for 30 min and then weighed; the difference between pre- and post-incubation weights was calculated as the volatility (g) of DEET formulations. The residual volume was expressed as the ratio to the pre-incubation weight of the absorbent cotton.

## Measurement of DEET concentration

The sample (50 μL) containing DEET was mixed with 100 μL of methanol containing 5 μg/mL butyl p-hydroxybenzoate (internal standard) in HPLC sample cup and set in LC-20AT HPLC system (Shimadzu Corp., Kyoto, Japan). Acetonitrile (30% v/v) was used as the mobile phase at a flow rate of 0.3 mL/min. DEET was separated using Inertsil® ODS-3 column (2.1 × 50 mm; GL Science Co., Inc., Tokyo, Japan) at 35°C, and detected at a wavelength of 210 nm using UV/VIS detector SPD-20A (Shimadzu Corp., Kyoto, Japan). In this study, the retention time of DEET was 6.25 min, and the subsequent favorable calibration curve was y = 1161.1x + 0.2069 ($R^2$ = 0.9976). The lower limit of quantification of DEET was 0.5 ng/mL.

## Partition coefficient of DEET formulations

Partition coefficient was measured using the shake-flask method based on the Organization of Economic Cooperation and Development (OECD) guidelines for the testing of chemicals 107 [24]. Briefly, 1-octanol (150 mL) and water (150 mL) were mixed, stirred overnight, and separated into 1-octanol and water phases. The separated 1-octanol and water were used in the

subsequent experiments. DEET (10% v/v) with or without 2% POE-POP was added to 45 mL of 1-octanol/water (1:1) mixture in a 50-mL test tube, stirred for 5 min at 22˚C, and centrifuged (800 g, 10 min, 22˚C). DEET concentration in each phase was then measured by HPLC, as described above. Similar operations were performed using 1-octanol/water (1:2 and 2:1 v/v) mixtures (total volume: 45 mL). Using DEET concentrations in 1-octanol ($C_o$) and water ($C_w$), logP was calculated using the following equation: LogP = $\log_{10} C_o/C_w$, and the average of the three values was expressed as logP in this study.

## Atomic force microscope (AFM) image

DEET-E and DEET-EPP were dissolved 10 times by volume in purified water and set on a rigid flat mica material, and AFM images were obtained using SPM-9700 (Shimadzu Corp., Kyoto, Japan). The AFM images of DEET formulations were created by combining phase and height images.

## Viscosity of DEET formulations

DEET formulations were incubated at 4, 25, and 50˚C, and their viscosity was measured using SV-1A (A&D Company, Limited, Tokyo, Japan). Viscosity was measured three times per measurement, and the average value was used.

## Degradability of DEET formulations

DEET formulations (5 mL each) were incubated at 4, 25, and 50˚C for one month in a 10-mL glass flask. Then, every 1, 2, 3, and 4 weeks, 100-μL formulation samples were withdrawn from the flask, and DEET concentrations were determined using HPLC, as described above.

## Measurement of POE-POP concentration

The POE-POP-containing sample was dried under reduced pressure. These residues were dissolved in 200 μL of formic acid (0.1% v/v) containing 1% (v/v) methanol, and liquid chromatography-analysis was carried out using an apparatus equipped with LPG-3400SD pump and Corona Veo detector (Thermo Fisher Scientific, Inc., Waltham, MA, USA). Separation was performed using TSK gel ODS-100S reverse-phase column (5 μm, 2.0 mm internal diameter × 150 mm length; Tosoh Co., Tokyo, Japan). The mobile phase consisted of 0.1% formic acid (solvent A) and methanol (solvent B); multi-step gradient elution was performed as follows: 1–85% solvent B from 0–2 min and 85–100% solvent B from 2–20 min. The analysis was conducted at approximately 23˚C and at a flow rate of 200 μL/min. The injection volume used was 20 μL. In this study, the retention time of POE-POP was 10.6 min, and the calibration curve of POE-POP was y = 0.5177x – 0.0685 ($R^2$ = 0.9996), and the lower limit of quantification of POE-POP was 0.13 μg/mL.

## POE-POP and DEET content in the skin

Wistar rats were used to measure the POE-POP and DEET content in the skin tissue according to our previous study [19]. Briefly, the abdominal hair of rats were removed. After one day, the rats were euthanized by injecting a lethal dose (200 mg/kg) of pentobarbital, and their skin tissue samples (2.01 $cm^2$) were collected and kept on a filter paper moistened with saline solution. Next, DEET formulation (3.36 μL, 1.67 μL/$cm^2$) was applied to the skin tissue sample, which was incubated at 37˚C for 4 h. The skin was then separated into four parts (skin surface, stratum corneum, epithelium, and dermis), homogenized in 300 μL of methanol, and the supernatant was collected by centrifugation (20,400 × $g$) at 4˚C for 20 min. In this study, the stratum

corneum was removed from the skin tissue using the tape-stripping method, and the samples were homogenized using purified water and methanol. Samples homogenized with purified water and methanol were used to measure the POE-POP and DEET content, respectively. The samples (200 μL) containing POE-POP and DEET were filtrated by 10 kDa ultrafiltration to remove high-molecular weight substances, and the POE-POP and DEET contents in the filtrated samples were measured as described above. Furthermore, the area under the skin concentration–time curve ($AUC_{0-8h}$) was determined using the trapezoidal rule.

## Digital and hematoxylin and eosin (H&E)-stained images of the rat skin

The abdominal hair of rats were removed. After one day, Wistar rats were treated with DEET formulations two times a day (9:00 and 17:00) for one month. Then, images of the surface of the rat skin tissues were captured using a digital camera. Subsequently, the rats were euthanized by injecting a lethal dose (200 mg/kg) of pentobarbital, and their skin tissue samples (2.01 cm$^2$) were collected and fixed in 10% formalin at 22˚C for 24 h. The fixed tissues were prepared in frozen blocks using the O.C.T. compound and liquid nitrogen, and 3-μm thick serial sections were prepared using Leica Cryostats CM3050S (Leica Biosystems Nussloch GmbH, Nussloch, Germany). Hematoxylin and eosin (H&E) staining was performed for morphological observation. The specimens were observed using Shimadzu BA210E (Shimadzu, Kyoto, Japan), and the central area of the abdominal skin treated with DEET formulations was photographed.

## Repellent effect of DEET

Fifty adult female *A. albopictus* were used to evaluate the repellent effect of DEET formulations, and the experiments were performed in 25×25×25 cm metal mesh cages (50 female mosquitoes per cage). A rubber glove with an approximately 5 × 5 cm section cut off was worn after treating the back of one hand with DEET formulation (DEET-E or DEET-EPP; 1.67 μL/cm$^2$), and the exposed area on the back of the hand was the treated area. A similar glove was worn on the other hand, with the exposed area on the back of the hand being the untreated one. The untreated area was placed in the cage, and the number of mosquitoes that landed on the exposed skin was counted for 5 min. The treated area was then placed in the cage, and the number of mosquitoes that landed on the exposed skin was counted for 5 min. These operations were repeated three times for multiple subjects, and the repellent rate was calculated using the following formula:

$$\text{Repellent rate } (\%) = \left( 1 - \frac{\text{number of landings in treated area}}{\text{number of landings in untreated area}} \right) \times 100$$

It is known that a minimum of 90% repellent rate is required for effective repellency. Taken together, the endpoint of the measurement was set at the point when the repellent rate showed below 90% in this study.

## Statistical analysis

Statistical analysis was performed using the JMP ver. 5.1 (SAS Institute). Student's *t*-test and ANOVA followed by Dunnett's multiple comparison were used for statistical analysis, and *p*-value < 0.05 was considered statistically significant. Data were expressed as the mean ± standard error (S.E.).

## Results

### Characteristics of DEET formulations with or without POE-POP

Stickiness, viscosity, and degradability are important factors that affect the feel and practical application of a skin formulation, and hydrophobicity (logP) is related to the skin penetration of DEET [6,12]. In addition, slow evaporation or volatility leads to the sustained repellent activity of insect repellents [6,12]. Therefore, we measured the above characteristics of DEET formulations as shown in Figs 1 and 2. The tensile stress and logP of DEET-E were $2.05 \times 10^{-2}$ N and 1.96, respectively, and 64.3% of DEET-E was volatilized after 30 min (Fig 1). The temperature was affected the viscosity and degradability of DEET. Therefore, we measured the changes in viscosity and degradability of DEET under the temperature range in daily life (4°C, 25°C, and 50°C). The viscosity of DEET-E decreased with increase in temperature from 4 to 50°C, and the degradability of DEET in DEET-E was approximately 4% at 4, 25, and 50°C (Fig 2). The tensile stress, volatile volume, and degradability of DEET-EPP were similar to those of DEET-E, although the logP value was significantly decreased by the addition of POE-POP. Moreover, POE-POP enhanced the viscosity of DEET formulations, and the production of colloidal particles by POE-POP addition was confirmed, as observed in AFM image (Fig 1D).

The experiments were performed at 25°C. n = 6. *$P<0.05$ *vs.* DEET for each category. Although the addition of 2% POE-POP did not affect the stickiness and volatile volume, the logP value of DEET decreased in addition to the formation of colloids.

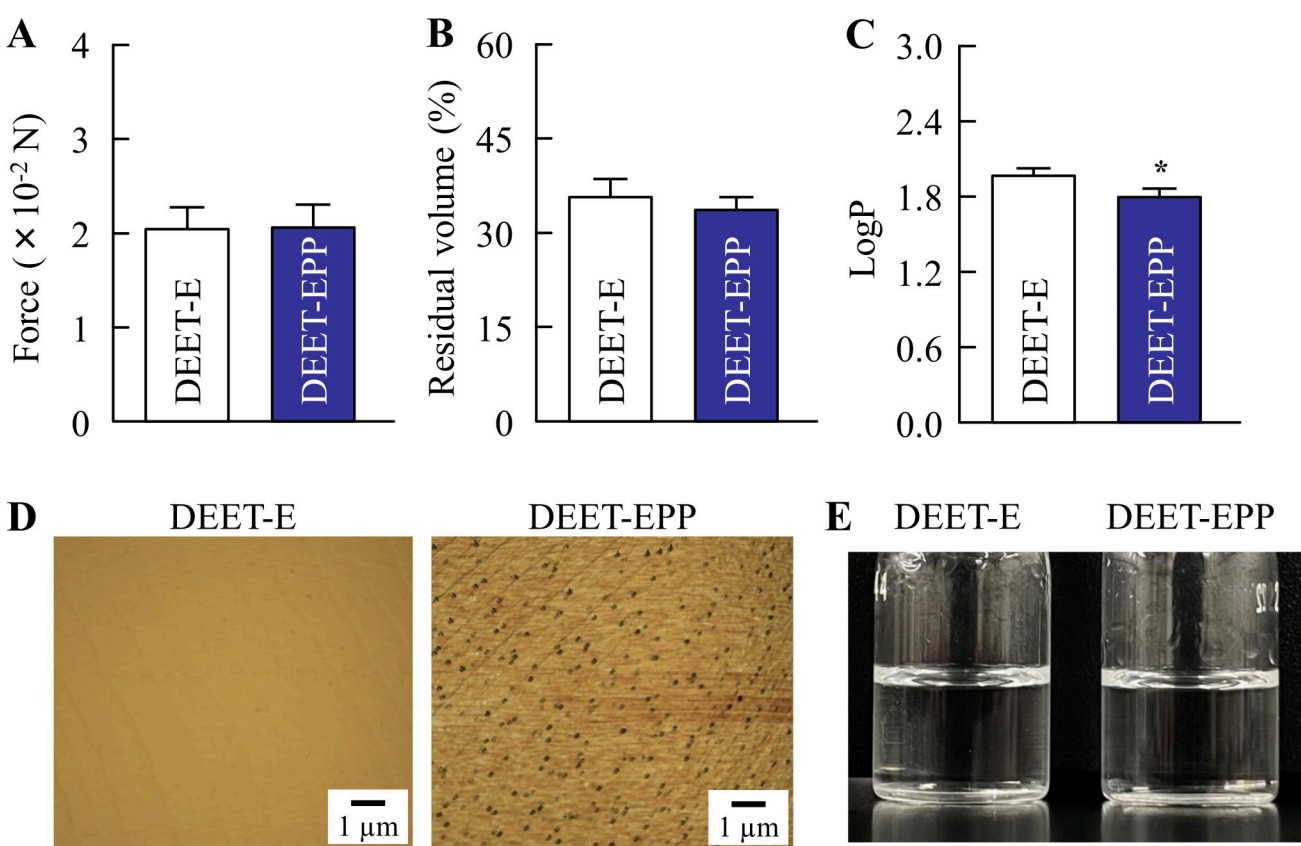

**Fig 1.** Differences in the tensile stress (stickiness) (A), volatile volume (B), logP (partition coefficient in octanol/water) (C), AFM image (D), and digital image (E) between DEET-E and DEET-EPP.

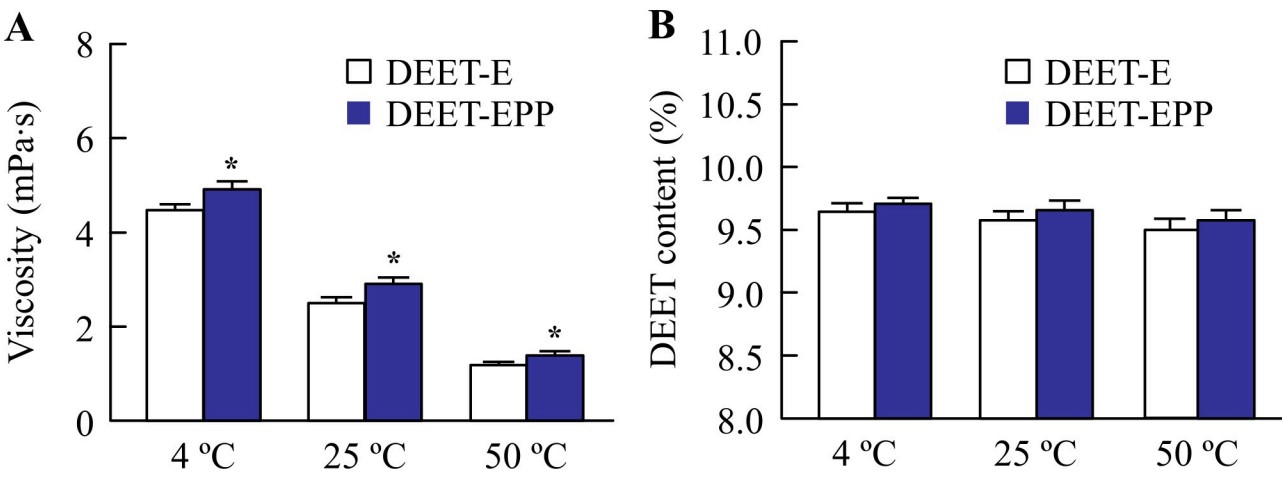

**Fig 2.** Effect of POE-POP addition on viscosity (A) and degradability (B) in DEET-E and DEET-EPP at 4–50°C.

DEET formulations were stored for one month. n = 8. *$P<0.05$ *vs*. DEET for each category. The viscosity of DEET-EPP was significantly higher than that of DEET-E. In contrast, no significant difference in DEET degradability was observed in DEET formulations with or without POE-POP at 4–50°C.

## Skin permeation behavior of POE-POP and DEET in the rats treated with DEET formulations

In the development of insect repellents, DEET retention on the skin surface increases the repellent action, whereas infiltration of DEET, additives, and solvent into the skin tissue causes toxicity. Therefore, we measured the skin permeation of POE-POP and DEET using the rat skin. Table 1 shows the POE-POP content in the skin of the rats treated with DEET-EPP. The POE-POP content was detected on the surface and in the stratum corneum and epithelium of the rat skin treated with DEET-EPP. The POE-POP content on the skin surface and in the stratum corneum was decreased, whereas the POE-POP content in the skin epithelium was increased over time. However, POE-POP was not detected in the dermis 4 h after treatment with DEET-EPP. Figs 3 and 4 show the behavior of DEET in the superficial layers (Fig 3) and tissues (Fig 4) of the rat skin treated with DEET formulations. Table 2 shows the $AUC_{0-8h}$ of DEET calculated from the data in Figs 3 and 4. DEET was detected in both superficial layers and tissue of the rat skin treated with DEET-E, and DEET levels in the superficial layers peaked 2 h after treatment, followed by a decrease. In contrast, DEET levels in the skin tissue (epithelium and dermis) gradually increased up to 8 h after treatment. The addition of POE-POP enhanced the retention of DEET in the superficial layers of the skin (surface and stratum corneum). In addition, the penetration of DEET into the skin tissue (epithelium and dermis) was

**Table 1. Changes in the POE-POP content in the skin tissue of rats treated with DEET-EPP.**

| | POE-POP content (µg/cm²) | | |
|---|---|---|---|
| | Surface and stratum corneum | Epithelium | Dermis |
| **2 h** | 1.05 ± 0.21 | 2.14 ± 0.20 | N.D. |
| **4 h** | 0.68 ± 0.10 | 2.55 ± 0.49 | N.D. |

N.D., not detectable. n = 5.

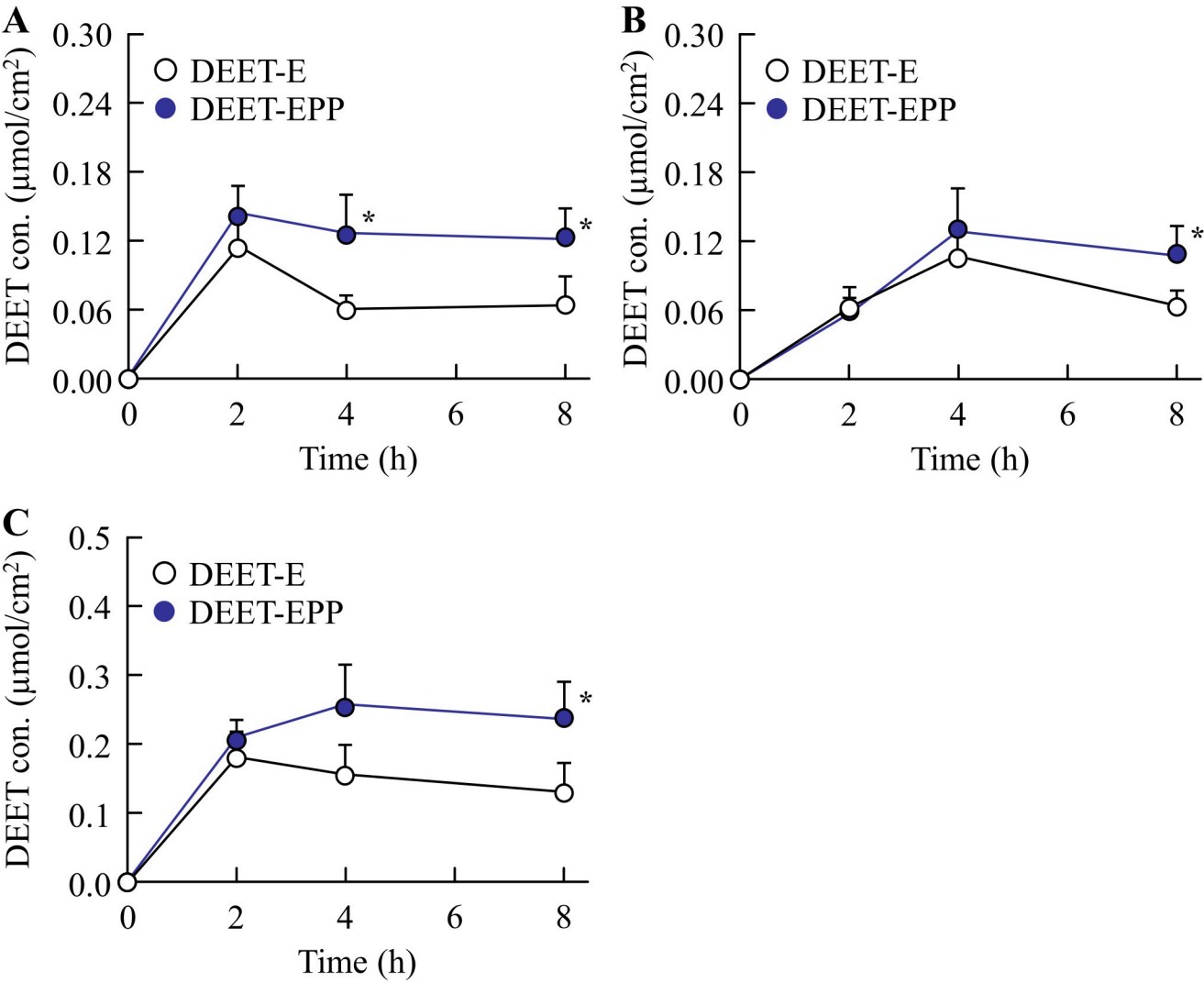

**Fig 3. Effect of POE-POP on the retention of DEET in the superficial layers of the skin treated with DEET formulations.** (A)–(C) Changes in the DEET content in the surface (A), stratum corneum (B), and superficial layers of the skin (surface and stratum corneum) (C). n = 6. *$P<0.05$ *vs*. DEET-E for each category. The addition of 2% POE-POP prolonged the retention of DEET in the superficial layers of the rat skin treated with DEET formulations.

significantly decreased by the addition of POE-POP. The $AUC_{0-8h}$ in the superficial layers and tissues of the rat skin treated with DEET-EPP was 1.44- and 0.84-fold of that of DEET-E, respectively (Table 2).

## Skin damage and repellent action of DEET formulations with or without POE-POP

We investigated whether the addition of POE-POP causes skin damage (toxicity). Fig 5 shows the skin conditions of rats treated with DEET-E and DEET-EPP for one month. Slight changes (redness) were visually observed in the rats treated with DEET-E, and the redness tended to decrease with the addition of POE-POP, as shown in the digital image. However, no difference was observed between the DEET-E- and DEET-EPP-treated rats with respect to H&E-stained skin images, and these images were similar to those of the untreated groups. Fig 6 shows the changes in the repellent action of DEET formulations with or without POE-POP. The repellent

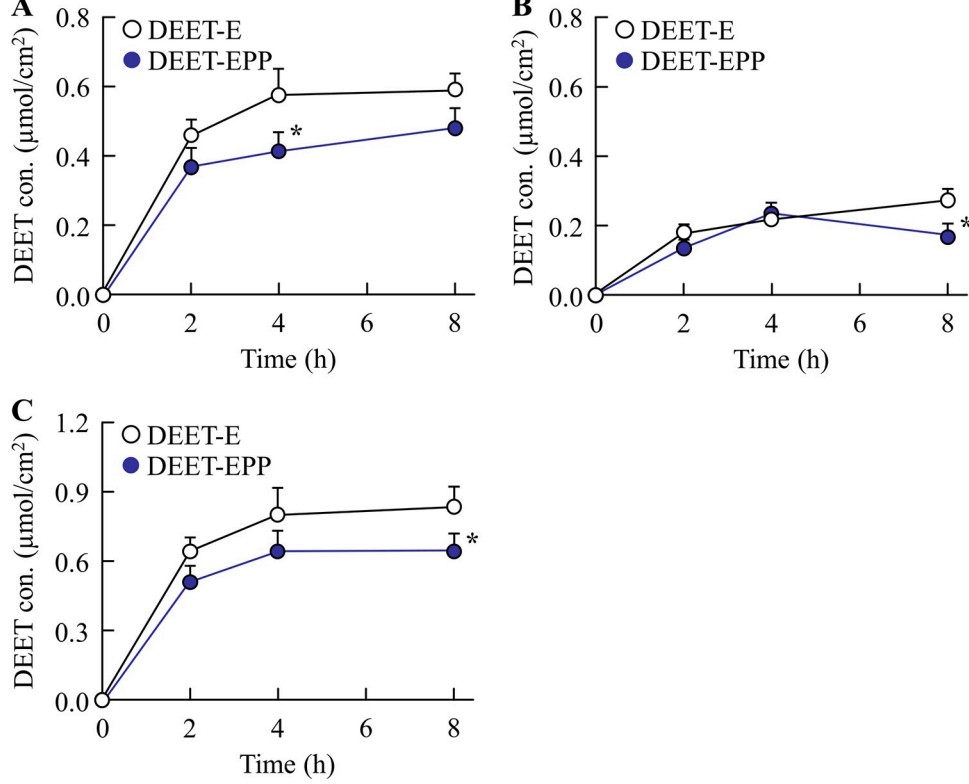

**Fig 4. Effect of POE-POP on the retention of DEET in the tissue of the skin treated with DEET formulations.** (A)–(C) Changes in the DEET content in the skin epithelium (A), dermis (B), and skin tissue (epithelium and dermis) (C). n = 6. *$P < 0.05$ *vs*. DEET-E for each category. The addition of 2% POE-POP decreased the skin permeation of DEET to the epithelium and dermis of the rat skin treated with DEET formulations.

effect of DEET against *A*. *albopictus* was better sustained by the addition of POE-POP. The repellent rate of DEET-E was less than 80% at 6 h after treatment. However, the repellent effect of DEET was prolonged by the addition of POE-POP, and the repellent rate of DEET-EPP was more than 90% at 12 h after treatment.

## Discussion

The desired characteristics of insect repellents are minimum permeation and sustained repellent activity in the superficial layers of the skin [12]. DEET, an insect repellent, is an oily volatile substance, and direct application of DEET may lead to systemic adverse effects and skin toxicity [9–11]. Therefore, solvents that can attenuate the skin penetration of DEET are required. In this study, we examined the usefulness of POE-POP addition to DEET and found

**Table 2. Changes in $AUC_{0\text{-}8h}$ in the skin tissue of rats treated with DEET formulations.**

|  | Superficial layers ($\mu$mol/cm$^2\cdot$h) | | | Skin tissue ($\mu$mol/cm$^2\cdot$h) | | |
|---|---|---|---|---|---|---|
|  | **Surface** | **Stratum corneum** | | **Epithelium** | **Dermis** | |
| **DEET-E** | 0.54 | 0.56 | 1.10 | 3.49 | 1.54 | 5.03 |
| **DEET-EPP** | 0.87 | 0.71 | 1.58 | 2.96 | 1.31 | 4.27 |

$AUC_{0-8h}$ was determined from data (mean) presented in Figs 3 and 4.

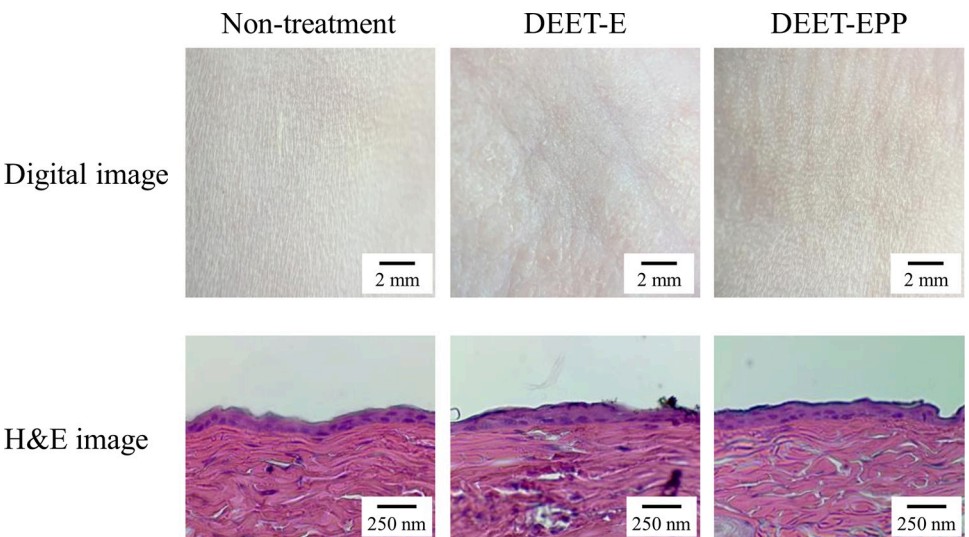

**Fig 5. Digital and H&E-stained skin images of the rats treated DEET formulations.** The rat skin was treated with DEET formulations two times a day (9:00 and 17:00) for one month. Slight redness in the skin of rats treated with DEET-EPP was visually observed to have decreased compared to those treated with DEET-E. In contrast, no skin damage was observed in H&E-stained images of the skin treated with DEET formulations with or without POE-POP.

that 2% POE-POP increased the skin retention and decreased the skin permeation of 10% DEET in 40% EtOH.

POP chains are used as modifiers for POE chains and hydrophobic base materials in the production of surfactants [25]. POE-POP, which is an amphiphilic random copolymer, has been used as a raw material in commercially available cosmetics. First, we demonstrated the

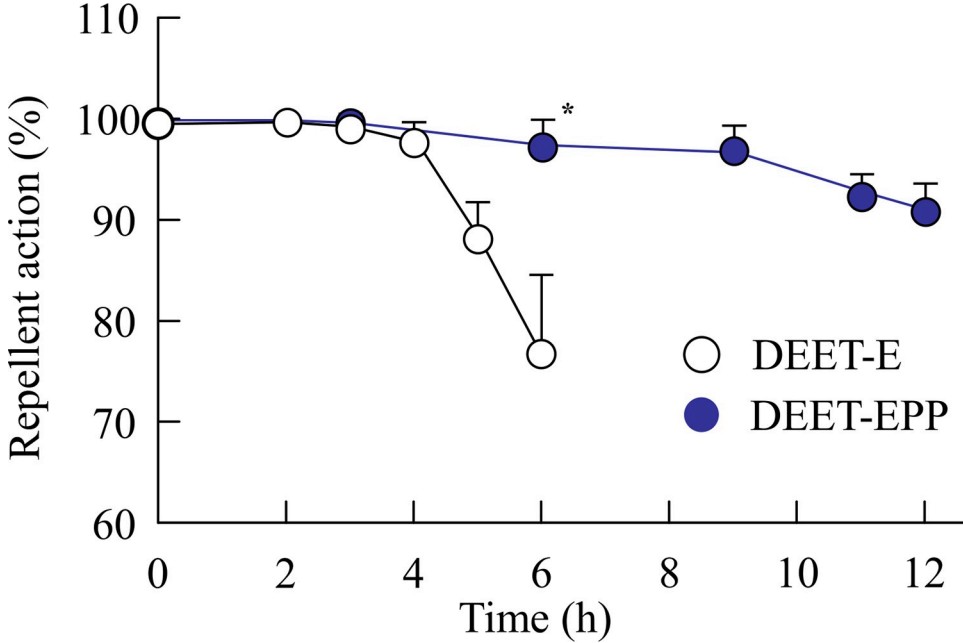

**Fig 6. Effect of POE-POP on repellent action of DEET formulations.** n = 3. *$P < 0.05$ *vs.* DEET for each category. The addition of POE-POP enhanced the repellent effect of DEET formulations.

effect of POE-POP on the characteristics of DEET formulations. DEET is highly soluble in alcohol but insoluble in water [6,7], and it is mainly dispersed in 40% ethanol. Therefore, 40% EtOH was used as the solvent for DEET in this study. DEET in EtOH (DEET-E) was a clear liquid with a viscosity of 2.49 mPa·s. Its degradability was 4.2% after one month of preparation at 25˚C (Fig 2), and the concentration of DEET was similar before and after one month. The residual volume after volatilization of DEET was 35.7% at 30 min after the start of the experiment (Fig 1B). Stickiness is a major factor affecting the feel of insect repellents and limits its practical applicability. The stickiness of DEET-E was $2.05 \times 10^{-2}$ N, which was similar to those of water ($1.91 \times 10^{-2}$ N) and vehicle ($1.98 \times 10^{-2}$ N). The stickiness, degradability, and volatility were similar between DEET formulations with and without POE-POP (Fig 1). Although POE-POP addition enhanced the viscosity of DEET-E (Fig 2) by 0.42 mPa·s at 25˚C, the increase was not noticeable at the time of use. These results showed that the addition of POE-POP did not affect the usability of the formulation. In contrast, DEET in DEET-E was soluble, whereas DEET in the solvent containing EtOH and POE-POP (DEET-EPP) showed colloidal dispersion (Fig 1D). It is known that the aggregations of a higher-order structure were formed by the introduction of a POP chain into the POE type surfactants controlled [26]. These results suggest that the addition of POE-POP may affect the affinity between the vehicle and DEET. Further studies are required to clarify this relationship.

Next, we investigated the effect of addition of POE-POP on the skin penetration of DEET formulations. The addition of POE-POP prolonged the retention of DEET in the superficial layers (skin surface and stratum corneum) and decreased the permeation of DEET into the skin tissue (epithelium and dermis) in rats treated with DEET-EPP (Figs 3 and 4). In general, it is known that the physicochemical properties such as molecular weight (MW) less than 500 Da and logP in the range of 1–4 characterize the drugs with high skin penetration [6,27–30]. Since the MW and logP of DEET are 191.274 g/mol and 1.96, respectively, DEET can pass through the cutaneous barrier and be absorbed *via* the deeper layers into the blood [8]. Furthermore, side effects of DEET, such as central nervous system toxicity, seizures, skin rash, and encephalopathy, have been reported [9–11]. Taken together, POE-POP decreased the lipophilic feature of DEET as the logP of DEET in DEET-EPP was significantly lower than that in DEET-E (Fig 1C), and the enhanced hydrophilicity of DEEF formulations attenuated their ability to penetrate into the skin, since skin surfaces such as stratum corneum are hydrophobic. Hence, a solvent consisting of EtOH and POE-POP may be able to reduce blood bioavailability of DEET, leading to development of a safe product. In addition, the formation of colloids by POE-POP may be attributed to the changes in logP (Fig 1D). Further studies are needed to elucidate the mechanisms underlying the decreased logP value, increased skin retention on the superficial layer, and drastic reduction in skin permeation of repellent.

Further, the safety and efficacy are important factors in the development of a new product. Therefore, we examined the toxicity of DEET formulations, with or without POE-POP, on the rat skin. The H&E-stained images of the rat skin treated with DEET-E and DEET-EPP were similar; however, a slight color change (redness) was observed on the surface of the rat skin treated with DEET formulations, and DEET-EPP showed a slightly less color change than that of DEET-E (Fig 5). The POE-POP were contained in various formulations, such as soap, shampoo, rinse, mouthwash, cosmetics et al., and no significant toxicity reported. These results supported the safety in these products used in everyday life. We also demonstrated the behavior of POE-POP in the rat skin treated with DEET-EPP (Table 1). POE-POP was detected in the superficial layers (skin surface and stratum corneum) and epithelium of the skin; however, it was not detected in the dermis 4 h after treatment with DEET-EPP. Moreover, the skin permeation of POE-POP was clearly less than toxic amounts, since the LD50 for oral administration of POE-POP (n = 4) was 1,800 mg/kg in the rat (Table 1). These results showed that

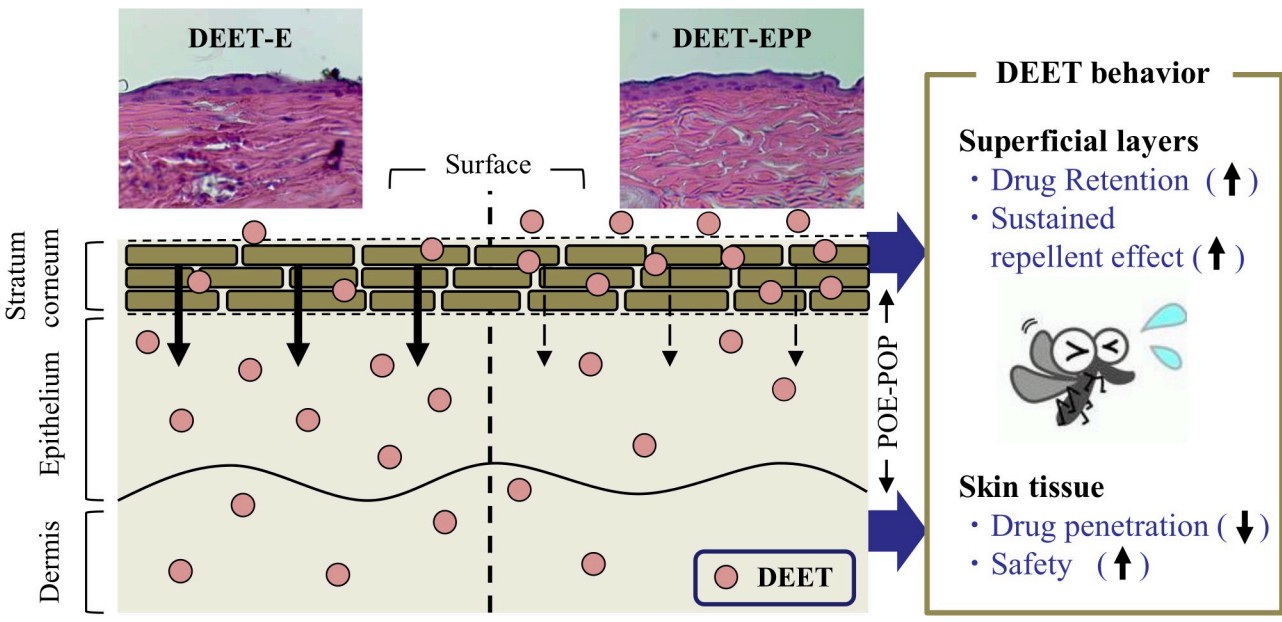

**Fig 7. Schematic representation of skin penetration and repellent action of DEET formulations with or without POE-POP.**

combination of POE-POP and EtOH may be used as a low-risk additive in insect repellent formulations.

Furthermore, we investigated the effect of POE-POP on the repellent activity of DEET (Fig 6). Since the selection of the mosquito model for evaluating the repellent activity is important, we used *A. albopictus* that is widely used in experiments with model animals [19]; the mosquito is highly active during the day and prefers to suck human blood. We evaluated the changes in the repellent activity of DEET formulations with or without POE-POP using *A. albopictus* with a long activity time. A repellent effect of more than 90% was observed up to 4 h after treatment with DEET-E, and the addition of 2% POE-POP prolonged the repellent effect of DEET. A repellent effect of 90% was observed for DEET-EPP up to 12 h (Fig 6). From these results, we hypothesized that POE-POP decreased the hydrophobicity of DEET in EtOH, and the decreased hydrophobicity attenuated the skin permeation and prolonged the retention of DEET in the superficial layers, leading to sustained repellent activity of DEET (Fig 7). It is known that an 8-h effect is the optimum for an insect repellent [6], and formulations containing 20–30% DEET have been used for long periods (approximately 6 h) [7]. In addition, penetrating the skin and avoiding entry into the bloodstream are added advantages contributing to the usefulness of DEET. From these findings, formulations consisting of low DEET content (10%), EtOH, and POE-POP may be useful as safe and effective insect repellent formulations. Nevertheless, it is important to investigate the detailed mechanism of the decrease in logP of DEET by the addition of POE-POP. Moreover, further studies are needed to demonstrate sustained repellent activity of DEET-EPP against other mosquitoes, including *Culex pipiens*, *Culex p. molestus* FORSKAL, and *Aedes aegypti*. Therefore, in future studies, we will aim to measure the synergistic repellent effects of DEET-E with POE and/or POP against other mosquitoes.

## Conclusions

We demonstrated that the use of a solvent consisting of 40% EtOH and 2% POE-POP reduced the skin permeation of DEET by decreasing its hydrophobicity and provided sustained

repellent activity compared with DEET-E formulation. This DEET formulation containing POE-POP was less tensile stress, and prolonged the retention of DEET in the superficial layers of the rat skin in comparison with DEET formulation containing cyclodextrins which we previous studied [19]. We conclude that the combination of EtOH and POE-POP is a safe and effective delivery system for repellent formulations containing DEET. The study findings reveal the prolonged activity of repellent formulations, which may be helpful in minimizing the health problems associated with DEET.

## Author Contributions

**Conceptualization:** Mayu Kawaguchi, Noriaki Nagai.

**Data curation:** Kana Matsumoto, Joji Yoshitomi, Hiroko Otake, Kanta Sato, Atsushi Taga, Tatsuji Sasabe, Kenji Nobuhara, Akira Matsubara.

**Formal analysis:** Mayu Kawaguchi, Kana Matsumoto, Joji Yoshitomi, Hiroko Otake, Kanta Sato, Atsushi Taga, Tatsuji Sasabe, Kenji Nobuhara, Akira Matsubara, Noriaki Nagai.

**Investigation:** Mayu Kawaguchi, Kana Matsumoto, Hiroko Otake, Atsushi Taga, Kenji Nobuhara, Noriaki Nagai.

**Methodology:** Mayu Kawaguchi, Tatsuji Sasabe, Kenji Nobuhara, Akira Matsubara, Noriaki Nagai.

**Project administration:** Akira Matsubara, Noriaki Nagai.

**Supervision:** Noriaki Nagai.

**Visualization:** Noriaki Nagai.

**Writing – original draft:** Noriaki Nagai.

**Writing – review & editing:** Noriaki Nagai.

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
