## [Decision Letter · Decision Letter 0]

22 Aug 2023

PONE-D-23-05194Poly(oxyethylene)/poly(oxypropylene) butyl ether prolongs the repellent effect of N,N-diethyl-3-toluamide on the skinPLOS ONE

Dear Dr. Nagai,

Thank you for submitting your manuscript to PLOS ONE. After careful consideration, we feel that it has merit but does not fully meet PLOS ONE’s publication criteria as it currently stands. Therefore, we invite you to submit a revised version of the manuscript that addresses the points raised during the review process.

We look forward to receiving your revised manuscript.

Kind regards,

Gopinath Ponnusamy Manogaran, PhD

Academic Editor

PLOS ONE

Journal Requirements:

Reviewers' comments:

Reviewer's Responses to Questions

**Comments to the Author**

1. Is the manuscript technically sound, and do the data support the conclusions?

Reviewer #1: Yes

Reviewer #2: Yes

2. Has the statistical analysis been performed appropriately and rigorously? 

Reviewer #1: Yes

Reviewer #2: Yes

3. Have the authors made all data underlying the findings in their manuscript fully available?

Reviewer #1: Yes

Reviewer #2: Yes

4. Is the manuscript presented in an intelligible fashion and written in standard English?

Reviewer #1: Yes

Reviewer #2: Yes

5. Review Comments to the Author

Reviewer #1: There is no problem with the research and its interpretation, but more detail on experimental method needs to be provided. Please respond to the comments provided below (minor revision).

1. In the drug efficacy evaluation experiment (in vivo), DEET-E results were stopped in the middle of the study. Author should add the reason for this.

2. Why different temperatures were used in viscosity and degradability of DEET formulation?

3. Why did you add ethanol to the formula. Author should indicate whether commercial DEET products also contain 40% ethanol.

4. Figure 2B, is there any reason to check the degradability after one month? Have you checked it at different days or week? What happens before and after one month? There should be any rationale to point out one specific time point.

5. Tensile stress is thought to be affected by temperature. How high was the temperature when measuring tensile stress?

6. In evaluating repellent effect of DEET, how large a space were female A. albopictus released into? Author should add this information.

Reviewer #2: Kawaguchi et al. showed that the addition of POE-POP attenuated the skin penetration and prolonged the repellent action of DEET. This study is interesting, and manuscript is well-written. But some modifications are needed before adoption. My comments are as follows

1. Please clarify which humane endpoints were established in your study.

2. Please provide ethical guidelines to justify the types and doses of anesthetic agents used for anesthesia and euthanasia of rat in your study.

3. Please indicate how your results compare with those from previous studies using cyclodextrin on this topic published in the literature.

4. The addition of POE-POPs increases hydrophilicity. Please explain more clearly why increased hydrophilicity would decrease skin penetration of DEET.

5. Addition of POE-POP formed the micelles. Is there any change in color? Please add photos of the formulation with and without POE-POP.

6. PLOS authors have the option to publish the peer review history of their article (what does this mean?). If published, this will include your full peer review and any attached files.

Reviewer #1: No

Reviewer #2: **Yes: **Tadatoshi Tanino

---

## [Author Response · Author response to Decision Letter 0]

25 Aug 2023

Response to Reviewer Comments

We carefully revised our manuscript according to the suggestions of the reviewers, and details are as follows.

< Q and A for Reviewer 1>

Q1. In the drug efficacy evaluation experiment (in vivo), DEET-E results were stopped in the middle of the study. Author should add the reason for this.

A1. Thank you very much for pointing this out. It is known that a minimum of 90% repellent rate is required for effective repellency. Taken together, the endpoint of the measurement was set at the point when the repellent rate showed below 90%. In order to respond to the reviewer’s comment, we added the information in the Materials and methods (line 231-233).

Q2. Why different temperatures were used in viscosity and degradability of DEET formulation?

A2. The reviewer’s comment is correct. The temperature was affected the viscosity and degradability of DEET. Therefore, we measured the changes in viscosity and degradability of DEET under the temperature range in daily life (4 °C, 25 °C, and 50 °C). In order to respond to the reviewer’s comment, we added the content in the Results (line 250-253).

Q3. Why did you add ethanol to the formula. Author should indicate whether commercial DEET products also contain 40% ethanol.

A3. It is mainly dispersed in 40% ethanol, since DEET does not have a high affinity for water. From these background, many commercial DEET products also use 40% ethanol as a solvent. In order to respond to the reviewer’s comment, we added the information in the Discussion (line 347).

 

Q4. Figure 2B, is there any reason to check the degradability after one month? Have you checked it at different days or week? What happens before and after one month? There should be any rationale to point out one specific time point.

A4. The reviewer’s comments are very important. In this study, we collected the samples every 1, 2, 3, and 4 weeks, and measured changes over time. The concentration of DEET was similar before and after one month. In order to respond to the reviewer’s comment, we added these contents in the Materials and methods, and Discussion (line 171, 349-350).

Q5. Tensile stress is thought to be affected by temperature. How high was the temperature when measuring tensile stress?

A5. Thank you very much for pointing this out. The measurement of tensile stress was performed at 22 °C. In order to respond to the reviewer’s comment, we added this information in the Materials and methods (line 123).

Q6. In evaluating repellent effect of DEET, how large a space were female A. albopictus released into? Author should add this information.

A6. The reviewer’s comment is correct. The evaluation of repellent effect of DEET was performed in 25×25×25 cm metal mesh cages (50 female mosquitoes per cage). In order to respond to the reviewer’s comment, we added this information in the Materials and methods (line 220-221).

Thank you for great comments. 

< Q and A for Reviewer 2>

Q1. Please clarify which humane endpoints were established in your study.

A1. Thank you very much for pointing this out. The death of a regulated animal as a likely outcome or planned experimental endpoint is not included in this study. Thank you for pointing out this.

Q2. Please provide ethical guidelines to justify the types and doses of anesthetic agents used for anesthesia and euthanasia of rat in your study.

A2. Thank you for pointing out this. The types and doses of anesthetic agents used for anesthesia and euthanasia of mice and rats were determined according to AVMA Guidelines for the Euthanasia of Animals: 2020 Edition. In order to respond to the editor’s comment, we added the information in the in the Materials and methods (line 97-99).

Q3. Please indicate how your results compare with those from previous studies using cyclodextrin on this topic published in the literature.

A3. The reviewer’s comments are very important. The DEET formulation containing POE-POP was less tensile stress, and prolonged the retention of DEET in the superficial layers of the rat skin in comparison with DEET formulation containing cyclodextrins. From these results, the addition of POE-POP is expected to lead to the development of good DEET formulations. In order to respond to the editor’s comment, we added the information in the in the Conclusions (line 424-426).

Q4. The addition of POE-POPs increases hydrophilicity. Please explain more clearly why increased hydrophilicity would decrease skin penetration of DEET.

A4. Thank you very much for pointing this out. Skin surfaces such as stratum corneum are hydrophobic, and the enhanced hydrophilicity of DEEF formulations attenuated their ability to penetrate into the skin. In order to respond to the reviewer’s comment, we added the information in the in the Discussion (line 374-376).

Q5. Addition of POE-POP formed the micelles. Is there any change in color? Please add photos of the formulation with and without POE-POP.

A5. The reviewer’s comment is correct. No change in color tone was observed regardless of addition or non-addition of POE-POP. In order to respond to the reviewer’s comment, we added photos of the formulation with and without POE-POP in the Figure 1E (Figure 1E).

Thank you for great comments.

---

## [Decision Letter · Decision Letter 1]

20 Sep 2023

Poly(oxyethylene)/poly(oxypropylene) butyl ether prolongs the repellent effect of N,N-diethyl-3-toluamide on the skin

PONE-D-23-05194R1

Dear Dr. Noriaki Nagai,

We’re pleased to inform you that your manuscript has been judged scientifically suitable for publication and will be formally accepted for publication once it meets all outstanding technical requirements.

Kind regards,

Gopinath Ponnusamy Manogaran, PhD

Academic Editor

PLOS ONE

Additional Editor Comments (optional):

Reviewers' comments:

Reviewer's Responses to Questions

**Comments to the Author**

1. If the authors have adequately addressed your comments raised in a previous round of review and you feel that this manuscript is now acceptable for publication, you may indicate that here to bypass the “Comments to the Author” section, enter your conflict of interest statement in the “Confidential to Editor” section, and submit your "Accept" recommendation.

Reviewer #1: All comments have been addressed

Reviewer #2: All comments have been addressed

2. Is the manuscript technically sound, and do the data support the conclusions?

Reviewer #1: Yes

Reviewer #2: Yes

3. Has the statistical analysis been performed appropriately and rigorously? 

Reviewer #1: Yes

Reviewer #2: Yes

4. Have the authors made all data underlying the findings in their manuscript fully available?

Reviewer #1: Yes

Reviewer #2: Yes

5. Is the manuscript presented in an intelligible fashion and written in standard English?

Reviewer #1: Yes

Reviewer #2: Yes

6. Review Comments to the Author

Reviewer #1: I have reviewed the revised manuscript.

The authors answered all questions.

So, I consider that it has been properly revised.

Reviewer #2: The authors completely described my all comments without problems. This revised manuscript is acceptable now.

7. PLOS authors have the option to publish the peer review history of their article (what does this mean?). If published, this will include your full peer review and any attached files.

Reviewer #1: No

Reviewer #2: No

---

## [Editor Report · Acceptance letter]

25 Sep 2023

PONE-D-23-05194R1 

Poly(oxyethylene)/poly(oxypropylene) butyl ether prolongs the repellent effect of N,N-diethyl-3-toluamide on the skin 

Dear Dr. Nagai:

I'm pleased to inform you that your manuscript has been deemed suitable for publication in PLOS ONE. Congratulations! Your manuscript is now with our production department. 

Kind regards, 

on behalf of

Dr. Gopinath Ponnusamy Manogaran 

Academic Editor

PLOS ONE